# Association of ABCG2 421G>T (rs2231142) Polymorphism with rosuvastatin induced adverse effects in dyslipidemic patients: Implication for personalized medicine

Mahjabeen Sharif[1]*, Kulsoom Farhat[1], Mudassar Noor[2], Ahsan Maqbool Ahmad[3], Dilshad Ahmed Khan[4], Raja Kamran Afzal[5], Muhammad Bilal Siddique[5]

1 Department of Pharmacology and Therapeutics, Army Medical College Rawalpindi, National University of Medical Sciences, Rawalpindi, Pakistan, 2 Department of Pharmacology and Therapeutics, CMH Institute of Medical Sciences, Bahawalpur, National University of Medical Sciences, Rawalpindi, Pakistan, 3 Premier Research and Innovative Management (PRIME), Islamabad, Pakistan, 4 Department of Pathology, National University of Medical Sciences, Rawalpindi, Pakistan, 5 Armed Forces Institute of Cardiology & National Institute of Heart Diseases, Rawalpindi, Pakistan

* mahjabeen30@hotmail.com

## Abstract

Statins are considered as the first line drugs for the treatment of hyperlipidemia. Despite proven efficacy of rosuvastatin, inter-individual variations in plasma rosuvastatin levels have been documented in various studies which causes variable response to statin tolerance. This study aims to evaluate the possible association of ABCG2 421 G>T (rs2231142) polymorphism with inter-individual variations in plasma rosuvastatin levels which potentially increases the rosuvastatin related adverse effects. This quasi experimental study was carried out from June 2022 till December 2023 in two tertiary care hospitals of Pakistan. Hyperlipidemic patients with low density lipoprotein more than 130 mg/dl were enrolled through non-probability purposive sampling. All the enrolled patients were treated with rosuvastatin 10 mg once daily for 12 weeks. Fasting lipid profile, serum creatine phosphokinase, liver and renal function tests were measured at the start of study and after 12 weeks of intervention with rosuvastatin. Blood samples were also collected for genotyping and determination of plasma rosuvastatin levels. Frequency of ABCG2 421 G>T polymorphism for wild type GG, heterozygous mutant GT and homozygous mutant TT genotypes were 54.5, 36.2 and 9.3% respectively. Minor allele frequency was 0.27. Patients with TT and GT genotypes have significantly raised plasma levels of rosuvastatin with mean value of 30.23±4.8 ng/mL and 22.35±5.1 ng/mL respectively as compared to wild GG genotypes 13.95±8.9 ng/mL ($p$=<0.001). Frequency of myopathy, hepatotoxicity and nephrotoxicity in study population was 5.3, 3.2 and 4.8% respectively. All the genetic models including co-dominant model GT (OR= 5.45, 95% CI: 3.09–9.62, $p$=<0.0001), TT (OR= 88.51, 95% CI: 24.84–315.44), dominant model (OR= 8.45,

**Data availability statement:** All relevant data are within the manuscript and its Supporting Information files.

**Funding:** The author(s) received no specific funding for this work.

**Competing interests:** The authors declare no competing interests.

95% CI: 4.91–14.52, *p*=<0.0001), recessive model (OR=37.29, 95%CI 11.06–125.78, p<0.001), over-dominant model, (OR= 2.26, 95% CI: 1.42–3.60, *p*=<0.0001) showed significant association with rosuvastatin adverse effects. It is inferred that patients having T variant allele is associated with higher plasma rosuvastatin concentration and increased the risk of development of adverse effects compared with G allele carriers. It is therefore suggested that genetic profiling may be done for dose tailoring to minimize the statin intolerance.

## Introduction

Cardiovascular diseases are the leading cause of morbidity and mortality worldwide. Elevated plasma low density lipoproteins (LDL-c) is the well established risk factor for the development of atherosclerosis [1]. Amongst all antihyperlipidemic drugs, statins are considered as the corner stone therapy for primary and secondary prevention of ischemic heart disease (IHD). Rosuvastatin due to its long half life and better efficacy, is commonly used antihyperlipidemic drug. It is highly efficacious in reducing LDL-c. A meta-analysis on 108 clinical trials involving 19,596 patients has revealed that 10 mg rosuvastatin decreases 46% LDL-c after 12 weeks of treatment. Apart from lipid lowering effect, rosuvastatin also exerts pleotropic effects on blood vessels including antioxidant property, reduction of inflammatory response and improvement of endothelial function [2].

Despite widespread therapeutic benefits on cardiovascular diseases, its use is associated with various adverse effects which leads to poor compliance. Discontinuation of rosuvastatin therapy due to statin related adverse effects leads to increased morbidity and mortality. Statin related muscular adverse effects are the most commonly observed side effects associated with treatment discontinuation [3]. Incidence of myopathy varies in different ethnic populations of world ranging from 5 to 20 percent. Although the exact mechanism of statin induced myopathy is yet to be elaucidated. Major risk factors for statin induced myopathy include female gender, genetics, advanced age, low body mass index, increased physical activity, and trauma [4]. These potential risk factors causes impaired protein prenylation, mitochondrial dysfunction, oxidative stress and apoptosis of myocytes. Studies have revealed a wide spectrum of muscular symptoms ranging from mild myalgias and myositis to myoglobinuria and life threatening rhabdomyolysis, while statin induced liver and kidney injury are scarce adverse effects [5]. There is an urgent need to explore the potential contributing factors for development of these side effects so as to provide alternative therapeutic strategies [6]. Studies have revealed that high rosuvastatin plasma levels increases the risk of adverse effects like development of myopathy, impaired liver and renal functions [7,8]. Being a hydrophilic drug, intestinal absorption and bioavailability of rosuvastatin is dependent on transporters like BCRP which is encoded by ABCG2 genes. BCRP limits intestinal absorption and increases the hepatic elimination of rosuvastatin. Non synonymous ABCG2 c. 421 G>T variant is present on exon 5 of ABCG2 gene in which 421G>T (rs2231142) transversion causes replacement

of amino acid glycine to lysine at codon 14, so altering the function of BCRP transporter. Studies have reported that this genetic variant is prevalent in Caucasians and Chinese population causes decreased efflux of rosuvastatin from intestines hence net absorption and bioavailability of rosuvastatin is increased which predisposes to increased risk of potential adverse effects [8,9].

Impact of genetic polymorphism of ABCG2 421 (rs2231142) on rosuvastatin pharmacokinetics and adverse effects has been studied in different populations [10] but data regarding influence of this genetic variability on rosuvastatin pharmacokinetics and adverse effects is yet to be determined in Pakistani population. Extrapolation of data from different population is always unreliable. This study aims to fill this gap and was undertaken to investigate prevalence of ABCG2 G > T (rs2231142) polymorphism and possible association of ABCG2 G > T variants with variable plasma rosuvastatin levels and increased incidence of myopathy, hepatotoxicity and renal toxicity in Pakistani population. Unveiling genetic determinant of rosuvastatin adverse effects in Pakistani cohort will help health care professionals to tailor and optimize the treatment regimen for hyperlipidemic patients. Findings of this study will pave the way for implementation of genetic screening before prescribing rosuvastatin for improving treatment outcome of hyperlipidemic patients.

## Materials and methods

### Ethical approval for study protocols

This quasi experimental study was carried out on three hundred and seventyfour patients from 28th June 2022–15th December 2023. Study adhered to the principles of Current Good Clinical Practices (cGCP) and Helsinki Declaration. Ethical approval was sought from Institutional Ethics Review Board of National University of Medical sciences to ensure the ethical norms (Reference letter no: PGHI-DMe(VR)-RCD-06–001). The study participants were furnished with a written informed consent that comprised of description of the study's protocols and an assurance of the preservation of data confidentiality.

### Characteristics of study population

Patients were selected through non probability purposive sampling. Following criteria was used for inclusion of study participants: (1) both male, female patients aged 30–70 years (2) elevated LDL-c levels more than 3.4 mmoles/L (3) have normal serum CPK levels (<190 U/L) (4) normal liver and renal function tests [11]. Exclusion criteria included (1) those Patients taking antihyperlipidemic drugs other than rosuvastatin,or any other drug known to interact with rosuvastatin (2) patients suffering from hepatic and renal diseases (3) patients with raised serum CPK levels >190 U/L (4) Pregnant and nursing females [8].

### Study protocol

Patients fulfilling the inclusion criteria were enrolled for this study. All the patients received rosuvastatin (10 mg) once a day for 12 weeks [12]. Study was carried out according to STROBE guidelines. Four thousand six hundred and thirty (4630) patients were screened for the

diagnosis of hyperlipidemia. Three thousand eight hundred and twenty (3820) patients were excluded as their LDL was not more than 3.4 mmoles/L, so did not fulfill the inclusion criteria. Moreover 40 patients did not agree to participate in this study. Total 810 patients were enrolled. Out of them 434 did not come for follow up visit after 12 weeks of treatment with rosuvastatin. Finally 376 patients completed 12 weeks therapy. Patient flow chart is shown in S1 Fig.

Compliance with prescribed rosuvastatin was ensured by telephonic calls. At day 0, socio-demographic profile of patients related to age, gender, ethnicity, weight, height, body mass index (BMI) was collected. 5 ml blood was drawn from each study participant for biochemical analysis which include serum creatine phosphokinase (CPK), liver function tests (LFTs), renal function tests (RFTs), Thyroid Stimulating Hormone (TSH) on day 0 and after 12 weeks of intervention with

rosuvastatin, 10 ml blood was drawn after 12 hours of administration of last dose for estimation of rosuvastatin plasma levels at steady state (through HPLC), biochemical analysis and for genotyping. To determine plasma levels of rosuvastatin, centrifugation of the blood of each study participant was done at 3000 revolutions per minute (rpm) for 10 minutes to separate plasma from blood cells and then plasma samples were stored at −80°C for further analysis [13].

## Criteria for assessment of severity of myopathy

Statin induced myopathy is defined as muscular weakness with creatine phosphokinase (CPK) level more than 10 times the upper normal limit. Severity of myopathy with rosuvastatin was assessed on the basis of following statin related myotoxicity (SRM) phenotype classification.

SRM 0: CPK < 4 X ULN (upper limit of normal) with no muscular symptoms. SRM 1. No CPK rise with mild muscular symptoms. SRM2: CPK < 4XULN with intolerable myalgias. SRM3: patientsts having CPK > 4 X UNL - < 10XUNL with myopathy. SRM4: CPK > 10XULN - < 50XULN with severe myopathy. SRM 5 & 6: patients with rhabdomyolysis and autoimmune mediated necrotizing myositis [14,15].

## Criteria for grading of acute kidney injury

Drug induced acute renal injury (AKI) is defined as increase in serum creatinine 1.5 times higher than upper normal limit. Criteria for assessing the severity of rosuvastatin induced AKI is as follows.

Grade1: Creatinine 1.5–2XUNL Grade2: Creatinine 2–3XUNL Grade 3: creatinine3XUNL or >3->4XUNL. Grade 4: Life threatening consequences & dialysis required [16].

**Criteria for grading of rosuvastatin induced liver injury** is as follows.

Statin induced liver injury is defined as an increase in alanine aminotransferase (ALT) 3 times higher than the upper normal limit (grade 2 liver injury is clinically significant). Criteria for assessing the severity of drug induced liver injury is as follows:

Grade 0: Total bilirubin within normal range, serum alanine transaminase (ALT) and alkaline phosphatase (ALP) <1.25XUNL Grade1: Bilirubin>1–1.5XULN, ALT and ALP values ranges between 1.25–2.5XULN Grade 2: Total bilirubin>1.5–2.5XULN, ALT and ALP > 2.5–5.0XULN. Grade 3: Total bilirubin >2.5–5XULN, ALT and AST > 5–10XULN Grade 4: Total bilirubin >5XULN, ALT and ALP > 10XULN [17].

## Analysis of blood and plasma samples of study participants

1. **Genotyping Analysis of ABCG2 421G >T (rs2231142).** For Polymerase Chain Reaction (PCR), genomic DNA was extracted from 200 µL of whole blood from each patient by using chelex method and was stored at -20ºC. Gel electrophoresis was performed to analyze and confirm the presence of DNA fragments. For genotyping of single nucleotide polymorphism of ABCG2 421 G > T (rs2231142) Amplification Refractory Mutation System- Polymerase Chain Reaction (TETRA-ARMS-PCR) was used. Four primers were designed for detection of alleles. Sequences of G and T allele specific primers (forward inner and reverse inner) are CCGAAGAGCTGCTGAGAATTG and TCTGACGGTGAGAGAAAACTCAA, having amplification product size of 246 base pairs (bp) and 200 bp respectively. Sequences of two common reverse and common forward primers are TATAGCAGGCTTTGCAGACATCTA and ATTTTATCCACACAGGGAAAGTCCTA, having amplification product size of 401 bp, were designed and synthesized. Reconstitution of all the four primers was done according to manufacturer's guidelines. Finally, 5 pmol/µl of primer mix was prepared. For detection of mutant alleles, 25µl of PCR reaction mixture was prepared in 0.2 ml PCR tubes by adding 01µl of primer mixture, 12.5µl of master mix (containing DNA polymerases, dNTPs, $MgCl_2$ and Taq buffer), 8.5 µl of PCR grade water and 3µl of extracted DNA. Gradient PCR (ThermoFisher, USA) was used to determine the best annealing temperature for each primer set and the final annealing temperature was optimized at 60°C. Optimized PCR program

consisted of initial denaturation at 95 °C for 4 min followed by 32 cycles of denaturation at 95 °C for 30 s, then annealing at 60°C for 45 s, then extension at 72 °C for 30 s and final extension at 72 °C for 5 min [13]. PCR product was subjected to gel electrophoresis using 2% agarose gel. 20µl of amplified DNA was loaded to agarose gel along with a 100 bp DNA marker (ThermoScientific, Lithuania). After electrophoresis, the DNA bands were visulized (Supplementary FigureS1) by Gel Documentation System (SYNGENE, USA) and the variant alleles were identified according to the size of DNA fragments [14] and were shown in supplementary S2 Fig.

**2 Analysis of rosuvastatin concentration in plasma.** Plasma rosuvastatin levels were determined by using Knauer K-1001 High Performance Liquid Chromatography (HPLC, Germany) connected with ultra violet (UV) detector. Plasma samples were first thawed at room temperature followed by vortexing for 2 minutes. Solid phase extraction (SPE) was done by using 3cc cartridges. 1 ml methanol was used for conditioning of cartridges. 500 µL of plasma sample was loaded on these conditioned cartridges. From SPE cartridges, rosuvastatin was eluted with 1 ml of eluent which consisted of 0.5% glacial acetic acid in methanol. Evaporation of eluent was done by using nitrogen gas. In 100 µL of mobile phase this residue was dissolved. These samples were analyzed for determination of rosuvastatin quantity in plasma by using HPLC [18]. Octa Decyl-Silica (ODS-3) C18 column (5µm x 4.6 x 250 mm, GL Sciences Inc, Japan) was used for all the experiments with flow rate of 0.8 mL/min for 30 minutes run time in isocratic mode. Various experiments were performed with multiple mobile ratios to optimize the conditions. Best separation was achieved with mobile phase of 40:60 ratio of water (with 0.2% formic acid) to acetonitrile. Detector wave length of 254 nm was used and injection volume of 10 µL was used for all the analytical processes. For rosuvastatin, limit of detection was 0.01 µg/mL with linearity range 0.02–2.5 µg/ml. All the experiments were performed at room temperature [19].

## Statistical analysis

**Data was analyzed by using Statistical Package for Social Sciences (SPSS) software version 26.** For quantitative data, mean±standard deviation (SD) were calculated. Qualitative data was represented as frequencies and percentages. For determination of allele and genotypic frequencies, Hardy Weinberg Equilibrium (HWE) calculator was used and $p$ values greater than 0.05 indicated that observed alleles were in agreement with HWE assumption. To detect significant difference between baseline and post-treatment values of biochemical parameters, paired t-tests were used. For grading of severity of rosuvastatin induced myopathy, hepatic and renal injury, chi square test was applied. Association of ABCG2 421 G > T genotypes with variable plasma rosuvastatin levels and biochemical parameters were analyzed by using One Way ANOVA followed by Post Hoc Tuckey Test. SNP Stat software was used to assess the association of ABCG2 421 G > T genotypes with risk factor for development of adverse effects and logistic regression analysis was used by calculating odd ratios (OR) and 95% confidence interval. $p$-value of ≤ 0.05 was considered as statistically significant.

## Results

### Demographic data and baseline characteristics of patients

Study included 376 patients out of which 215 (57.2%) were male and 161 (42.8%) were females with mean age 54.34±9.65 years (range 31 to 70 years). The mean body mass index (BMI) was 29.04±4.2 kg/m². Baseline characteristics of study population are shown in S2 Table.

### Genotype and allele frequencies of ABCG2421 G > T (rs2231142)

Our results showed that 205 (54.5%) patients have wild GG genotype, while 136 (36.2%) patients have heterozygous mutant genotype GT and 35 (9.3%) patients have homozygous mutant genotype TT. The minor allele (T) frequency (MAF) was 0.27. $p$ value of 0.07 indicated that observed frequencies were in agreement with HWE assumption. Genotype and allelic frequencies are shown in S2 Table and S2 Fig.

## Effect of ABCG2 421 G > T polymorphism on plasma rosuvastatin concentration

After 12 weeks of rosuvastatin therapy, mean plasma rosuvastatin concentration was 18.5 ± 9.18 ng/mL. Mean plasma rosuvastatin levels in patients carrying GG, GT and TT genotypes were 13.95 ± 8.9, 22.35 ± 5.1 and 30.23 ± 4.8 ng/mL respectively (S3 Fig). There was statistically significant difference amongst the mean plasma rosuvastatin levels of patients carrying GG, GT and TT genotypes. Patients having mutant homozygous genotypes TT had the highest plasma rosuvastatin concentrations (Table 1).

## Impact of ABCG2 421 G > T polymorphism on Statins-Associated Muscle Symptoms

Serum creatinine phosphokinase levels were significantly raised from 100.16 ± 26.28 to 272.42 ± 477.47 U/L after 12 weeks of intervention with rosuvastatin. Although there was statistically significant rise in CPK levels when compared between pretreatment and post treatment values (S3 Table), but in 294 (78.2%) patients CPK levels remained within the normal range (Table 3). Comparison of % age rise in three different genotypes GG, GT and TT after 12 weeks of rosuvastatin intervention, was statistically significant ($p = < 0.001$). Greatest percentage increase in CPK levels were observed in patients of TT genotypes and is shown in Table 2.

A wide spectrum of muscular symptoms were observed. 294 (78.2%) patients did not experience any pain and their CPK level remains within the normal range < 190 U/L. 16 (4.3%) patients did not have any muscular symptoms but their CPK levels were elevated less than 4 times upper normal limit (SRM 0). 11 (2.9%) patients experienced tolerable muscular pain but their CPK levels remained within the normal range (SRM1). 11 (2.9%) patients have muscular pain with CPK levels more than 4 times ULN (SRM2). 20 (5.3%) patients experienced myopathy and their CPK levels were more than 4 times ULN (SRM3). 20 (5.3%) patients have CPK levels more than 10 times ULN and experienced severe myopathy (SRM4), shown in Table 3.

Most of the patients having wild GG genotypes either did not experience muscular pain (189 patients, 50.3%) or mild symptoms were observed. SRM4 is considered as clinically significant myopathy and most of the patients with SRM4 were having TT (10 patients, 2.1%) or GT genotypes (5 patients, 1.3%) as shown in Table 3.

## Impact of ABCG2 421 G > T polymorphism on liver function tests

Before enrollment, all the study patients had normal LFTs. After 12 weeks of therapy, there was minor rise in total bilirubin, ALT and ALP but in most of the patients liver enzymes remained within the normal range. Mean difference between

**Table 1. Multiple comparisons of mean plasma concentrations of rosuvastatin with respect to different genotypes of ABCG2 421 G > T (rs2231142).**

| ABCG2 G > T genotypes | Plasma rosuvastatin concentration Mean±SD (ng/mL) | p-value | Comparison of two genotypes of ABCG2 G > T (rs2231142) | Mean difference±SEM of Cmax between two genotypes | p-value |
|---|---|---|---|---|---|
| GG | 13.95 ± 8.9 | <0.001* | GG> | 8.39 ± 0.82 | <0.001* |
| | | | GG&TT | 16.27 ± 1.36 | <0.001* |
| GT | 22.35 ± 5.1 | | GT&GG | 8.38 ± 0.82 | <0.001* |
| | | | GT&TT | 7.88 ± 1.4 | <0.001* |
| TT | 30.23 ± 4.8 | | TT&GG | 16.27 ± 1.36 | <0.001* |
| | | | TT> | 7.88 ± 1.4 | <0.001* |

Statistical analysis was performed by one way ANOVA followed by Post Hoc Tuckey test. p-value was > 0.05.

*= $p < 0.05$.

**Table 2. Association of ABCG2 421 G>T genotypes with %age changes (between pretreatment and post-treatment values) in biochemical parameters after 12 weeks of intervention with rosuvastatin 10 mg.**

| Parameters (Percentage change between pretreatment and post treatment values) | GG | GT | TT | p-value amongst GG, GT and TT | p value GG vs GT | p value GG vs TT | p value GT vs TT |
|---|---|---|---|---|---|---|---|
| %age change in CPK | 59.1±310 | 211.5±497.5 | 515.80±611.59 | <0.001* | 0.003* | <0.001* | <0.001* |
| %age change in total bilirubin | 36.7±76.8 | 39.15±98.6 | 88.29±144.68 | 0.009* | 0.970# | 0.007* | 0.015* |
| %age change in ALT | 19.39±63.9 | 62.7±93.7 | 140.39±219.7 | <0.001* | <0.001* | <0.001* | <0.001* |
| %age change in ALP | 21.78±53.6 | 51.9±111.51 | 89.46±136.39 | <0.001* | 0.006* | <0.001* | 0.065# |
| %age change in S.urea | 12.8±28.4 | 22.41±51.79 | 44.24±96.39 | <0.001* | 0.166# | <0.001* | 0.042* |
| %age change in S. creatinine | 14.47±49 | 24.9±61.13 | 58.9±115.8 | <0.001* | 0.281# | <0.001* | 0.012* |
| %age change in S. sodium | 0.82±2.55 | 1..09±2.5 | 3.35±1.5 | <0.001* | <0.001* | <0.001* | <0.001* |
| %age change in S. Potassium | 4.89±10.8 | 5.06±15.1 | 16.86±15.57 | <0.001* | <0.001* | <0.001* | <0.001* |

Data were expressed as mean±SD.

Statistical analysis was performed by one way ANOVA.

*=$p<0.05$.

NS=$p>0.05$.

**Table 3. Frequency of muscular symptoms in three different genotype groups of ABCG2 421 G>T.**

| Genotypes of ABCG2 34C>T | Grading criteria for rosuvastatin related myotoxicity | | | | | | | | p-value |
|---|---|---|---|---|---|---|---|---|---|
| | No of patients with no muscular symptoms & normal CPK levels | | No of patients with SRM0 | No of patients with SRM1 | No of patients with SRM2 | No of patients with SRM3 | No of patients with SRM4 | Total No of patients | |
| GG | 189 (50.3%) | | 6 (1.6%) | 3 (0.8%) | 2 (0.5%) | 0 (0%) | 5 (1.3%) | 205 (54.5%) | <0.001* |
| GT | 91 (24.2%) | | 10 (2.7%) | 8 (2.1%) | 9 (2.3%) | 13 (3.5%) | 5 (1.3%) | 136 (36.2%) | |
| TT | 14 (3.72%) | | 0 (0%) | 0 (0%) | 0 (0%) | 11 (2.9%) | 10 (2.6%) | 35 (9.3%) | |
| Total | 294 (78.2%) | | 16 (4.3%) | 11 (2.9%) | 11 (2.9%) | 24 (6.4%) | 20 (5.3%) | 376 (100%) | |

Statistical analysis was performed by chi square test.

*= Statistically significant results $p<0.05$.

pretreatment and post treatment values for total bilirubin, serum ALT and serum ALP were statistically significant ($p$=<0.001) and are shown in S3 Table.

Percentage increase in total bilirubin, ALT and ALP in patients of 3 different genotypes groups with their comparisons are shown in Table 5. During second visit, out of 376 patients, 326 (86.7%) patients had all the liver enzymes within the normal range. 09 (2.4%) study participants having GT genotype have grade 1(bilirubin >1–1.5XULN, ALT & ALP = 1.25–2.5XULN) and 04 (1.1%) patients have grade 2 (total bilirubin > 1.5–2.5XULN, ALT & ALP > 2.5–5.0XULN) rosuvastatin induced liver injury. 07 (1.86%) patients having TT genotype have grade 1 liver injury and 08 patients (2.1%) have grade 2 liver injury. While majority of the patients having GG wild type genotype have normal liver enzymes (Table 4).

**Table 4. Comparisons of grading of study participants for rosuvastatin induced liver injury in three different genotype groups of ABCG2 421 G>T polymorphism.**

| Genotypes of ABCG2 4 21G>T | Grading criteria for rosuvastatin induced hepatic injury | | | | | p-value |
| --- | --- | --- | --- | --- | --- | --- |
| | No of patients with normal LFTs | No of patients with Grade 0 hepatic injury | No of patients with Grade1 hepatic injury | No of patients with Grade 2 hepatic injury | Total no of patients | |
| GG | 198 (52.6%) | 6 (1.6%) | 1 (0.26%) | 0 (0%) | 205 (54.5%) | <0.001* |
| GT | 113 (30%) | 10 (2.6%) | 9 (2.4%) | 4 (1.1%) | 136 (36.1%) | |
| TT | 15 (4%) | 5 (1.3%) | 7 (1.86%) | 8 (2.1%) | 35 (9.3%) | |
| Total | 326 (86.7%) | 21 (5.6%) | 17 (4.52) | 12 (3.2%) | 376 (100%) | |

Statistical analysis was performed by chi square test.

*= Statistically significant results p <0.05.

There was minor rise in liver function tests which was clinically insignificant because after 12 weeks of treatment, rise in these enzymes was within the normal range except for 12 (3.2%) patients for which ALT was elevated 3 times more than the normal limits (Table 4).

## Impact of ABCG2 421 G>T polymorphism on renal function tests

After 12 weeks of therapy, there was a minor rise in serum urea, creatinine, sodium and potassium and 358 (95.2%) study patients, have renal function tests within normal range. Mean ±SD of baseline and post-treatment values of urea, creatinine, serum sodium and serum potassium were increased from 25.13±6.34 to 29.22±11.84 mg/dL, 0.86±0.34 to 1.0±0.48 mg/dL, 138.39±2.4 to 138.39±2.9 mmol/dL and 4.17±0.42 to 4.17±0.52 mmol/dL respectively as shown in Table 4. Comparison of percentage increase of all parameters of renal function tests amongst three different genotypes groups are shown in Table 5. 08 (2.1%) patients having GT genotype while 04 (1.1%) patients having TT genotypes suffered from grade 1 acute kidney injury while 01 (0.27%) patient having GT genotypes and 04 (1.1%) patients having TT genotypes suffered from grade 2 acute kidney injury in which serum creatinine levels were elevated 2–3 times the upper limit of normal. None of the patients have grade 3 or grade 4 acute kidney injury (Table 5).

**Table 5. Comparisons of grading of study participants for rosuvastatin induced acute kidney injury in three different genotype groups of ABCG2 421 G>T polymorphism.**

| ABCG2 4 21G>T | Grading of patients for rosuvastatin induced kidney injury | | | | p-value |
| --- | --- | --- | --- | --- | --- |
| | No of patients with Normal RFTs | No of patients with Grade1 AKI | No of patients with Grade 2 AKI | Total no of patients | |
| GG | 204 (54.3%) | 1 (0.27%) | 0 (0%) | 205 (54.5%) | <0.001* |
| GT | 127 (33.8%) | 8 (2.1%) | 1 (0.27%) | 136 (36.2%) | |
| TT | 27 (7.2%) | 4 (1.1%) | 4 (1.1%) | 35 (9.3%) | |
| Total | 358 (95.2%) | 13 (3.5%) | 5 (1.3%) | 376 (100%) | |

Statistical analysis was performed by chi square test.

*= Statistically significant results p <0.05.

## Genetic Model association of rs2231142 with rosuvastatin adverse effects

Genetic association between rs2231142 variants and rosuvastatin adverse effects were observed under different genetic models. Binary logistic regression analysis was applied and potential confounders like age, gender, BMI, WC, ethnicities, CPK, LFTs, RFTs were adjusted. All the genetic models showed significant association with rosuvastatin adverse effects. Under co-dominant model GT has 5.45 times (95% CI: 3.09–9.62, $p=<0.0001$) more chances of developing adverse effects with rosuvastatin when compared with wild type GG genotype. TT genotype has 88.51 (95% CI: 24.84–315.44) times more chances of developing rosuvastatin adverse effects compared with reference genotype GG. Under dominant model, GT/TT has 8.45 (95% CI: 4.91–14.52, $p=<0.0001$) times more probability of responding to rosuvastatin when compared with wild type GG genotype. In over-dominant model, genotype GT has 2.26 (95% CI: 1.42–3.60, $p=<0.0001$) times increased chances of developing adverse effects when compared with G/G-TT genotypes (S4 Table).

## Discussion

This is the pioneer study which for the first time revealed significant association of *ABCG2* 421G>T polymorphism with raised rosuvastatin plasma levels and rosuvastatin induced intolerance in Pakistani population. Study participants carrying homozygous mutant ABCG2 G>T TT and heterozygous mutant GT genotypes showed significantly higher plasma rosuvastatin levels subsequently causing relatively increased incidence of adverse effects as compared to patients with wild type GG genotype.

BMI of our study participants indicated that most of them were overweight and obese.

This finding is in line with previous studies which demonstrated that increased BMI and obesity are directly linked with hyperlipidemia as it causes altered lipid metabolism thus elevating the levels of LDL-c, VLDL-c, TGs, while decreasing HDL-c. Increased fatty tissue also stimulates the release of interleukin 6 (IL-6) and tumor necrosis factor α (TNF-α) which enhances insulin resistance and lipogenesis [20].

This study reported that genotype frequencies of ABCG2 G>T (rs2231142) were mostly of wild type GG (54.5 percent) while percentage frequency of heterozygous mutant GT and homozygous mutant TT genotype were 36.2 and 9.3 percent respectively. This genetic variant was also studied in Polish population and frequency of wild, heterozygous and homozygous genotypes were found to be 78, 20 and 2 percent respectively and minor allele frequency (MAF) was 0.12 percent [21]. Lonnberg and his co-workers found that in Finnish population, percentage frequencies of wild, heterozygous and homozygous mutant genotypes were 86.3, 0.7 and 13 percent respectively and MAF was 0.07 percent [22]. In line with these results, another study was conducted on Chinese population in which frequency of wild, heterozygous and homozygous mutant genotypes were 30.8, 59 and 10.3 percent respectively and frequency of reference G and mutant T alleles were 60.3 and 39.7 percent respectively [19].

Variable plasma drug levels were observed in study participants after 12 weeks of intervention. Highest rosuvastatin plasma concentration (Cmax) was observed in patients carrying TT genotypes followed by GT and GG carriers. This inter-individual variability in mean rosuvastatin plasma levels may be due to three different genotype groups of ABCG2 421 G>T which alters the function of BCRP efflux transporter leading to variable bioavailability of rosuvastatin [23]. Our findings are in accordance with results of a study in which Cmax of rosuvastatin was highest in patients with homozygous mutant genotype followed by heterozygous mutant and wild type genotypes [16]. Our results are also in line with previous clinical findings in which Cmax of rosuvastatin was found to be higher in patients carrying mutant allele of ABCG2 G>T (rs2231142) compared with those patients carrying wild alleles [24].

Inter individual variability in mean post treatment CPK levels in three genotype groups has been observed and patients carrying TT genotypes produced significantly higher levels of CPK compared with GT and GG genotypes. This variability in CPK levels might be due to genetic variant of ABCG2 421 G>T which decreases the function of BCRP efflux transporter leading to increased bioavailability and ultimately raised concentration of rosuvastatin in muscle and variable serum CPK levels. Higher levels of post treatment CPK levels in patients with TT genotype

predisposes to increased risk of myopathy. This may be one of the reason that SRM3 and SRM4 is mostly observed in patient with TT genotpes [25].

Wide spectrum of muscular symptoms were observed after 12 weeks of rosuvastatin treatment. Most of the study patients did not experience any muscular symptoms while some suffered from severe myopathy. Criteria for severe myopathy is defined as when CPK levels rises 10 times the upper normal limit with severe myalgias [25]. SRM4 is considered as clinically significant myopathy and most of the patients with SRM4 were having either TT and GT genotypes. Our results showed that possible underlying mechanism of myopathy in variant allele carriers T may be due to genetic polymorphism of ABCG2 421 G > T which leads to decreased function of BCRP efflux transporter which ultimately increases the rosuvastatin intestinal absorption leading to increased plasma concentration which predisposes to myopathy [26]. Our results were in accordance with findings of a study in which ABCG2 G > T is significantly associated with increased incidence of muscular adverse effects and variant allele has 2.1 times more chances to develop muscular adverse effects as compared to reference allele. Incidence of mild to moderate myalgias in this study were found to be 40.5% [27]. Contrary to our findings, is a study conducted by Lonnerg and his colleagues who reported that 17.4% of study participants experienced muscular symptoms but these muscular adverse effects were not associated with ABCG2 421G > T polymorphism [22].

Incidence of myopathy varies in different ethnic populations. In our study, incidence of myopathy was 5.3%. A study conducted on Jordanian population revealed dose dependent statin induced muscular symptoms as incidence of myalgias with 10 and 20 mg rosuvastatin was found to be 10.8 and 14.6% respectively [28]. In another study, frequency of myalgias in Pakistani population with rosuvastatin, atorvastatin and simvastatin were 3.2, 11.3 and 27.5% respectively. Higher incidence of myopathy with other statin compared with rosuvastatin may be due to hydrophilicity of rosuvastatin and hence less permeation in muscles [29]. A study conducted by Zakria and his co-workers revealed slightly higher frequency of myalgias in which 23% population developed muscular symptoms after receiving 10 mg of rosuvastatin [30].

Inter-individual variations in post treatment values of ALT and AST has been observed in 3 different genotype groups. Highest percentage rise in ALT and AST was observed in patients with TT and GT genotypes compared with GG carriers. As Cmax of TT and GT carriers was higher compared with GG carriers, so increased levels of rosuvastatin in liver causes altered lipid content of cell membrane of hepatocytes leading to increased leakage of liver enzymes in blood mainly in patients of TT and GT carriers.

According to latest guidelines of American College of Gastroenterology, drug induced hepatic injury is considered as significant only if ALT rises 3 times the upper limit of normal [31,32].

`Only 3.2% of study patients mostly carrying TT and GT genotypes have ALT levels more than 3 times the upper normal limit (grade 2 liver injury). Raised levels of LFTs may be due to alteration in lipid content of cell membrane of hepatocytes leading to increased permeability and leakage of liver enzymes [33]. Our observations were in accordance with a study in which patients with variant genotypes of ABCG2 G > T have 2.24 times increased risk of developing hepatic injury compared to reference genotype and incidence of acute hepatic injury was 7.9% [27]. In line with these findings, is another study in which only 1% of study participants had ALT levels more than 3XUNL with rosuvastatin therapy [33]. Results of our study were further corroborated with a study conducted by Aleem and his co-workers who reported that after 6 months of rosuvastatin therapy with 5 mg and 10 mg rosuvastatin, no significant rise in liver enzymes were observed and rosuvastatin was well tolerated by most of the patients [34].

Frequency of rosuvastatin induced acute renal injury in this study was found to be 4.8% which was mostly observed in patients with TT and GT genotypes. Drug induced acute renal injury becomes clinically significant when creatinine rises 1.5XULN [35]. Most of the previous studies reveals little or no significant renal adverse effects with rosuvastatin but in few rare studies rhabdomyolysis was developed with very high doses and prolong use of rosuvastatin [36]. A case report revealed that when 80 mg rosuvastatin was administered for 14 months in a patient, all renal function tests were deranged, and mild hematuria and albuminurea was also observed. All these changes were reversed with discontinuation

of rosuvastatin. Author reported that during rosuvastatin biotransformation, reactive oxygen species (ROS) and oxidative stress is produced which leads to minor elevation of liver and renal enzymes [37].

To our knowledge, this was the first study on Pakistani patients which demonstrated that genetic polymorphism of ABCG2 421G>T might alter the functions of BCRP efflux transporter. Patients carrying variant allele T have enhanced plasma rosuvastatin levels and slightly higher incidence of myopathy, hepatic and renal adverse effects with 10 mg rosuvastatin. It is therefore suggested that ABCG2 421 G>T might be a possible predictor for rosuvastatin intolerance but physician should not deter from prescribing rosuvastatin. To further mitigate the risk of rosuvastatin induced potential adverse effects in patients carrying T allele, low dose 5 mg rosuvastatin may be prescribed for its life long use for secondary prevention of cardiovascular diseases. The results of this study might improve tolerability of rosuvastatin by identifying genetic markers of adverse effects of rosuvastatin and pave the way for clinical implication of personalized treatment of hyperlipidemia. But further studies are warranted on other populations and different ethnicities to validate the findings of this study.

The current study has certain limitations as only Pakistani subjects were recruited in this research study which could limit the generalization of our findings on other populations. Furthermore impact of only one rosuvastatin transporter SNP was investigated which may overlook other potential genetic variations that could influence rosuvastatin treatment outcomes. There was no control group in this study which might introduce response bias in clinical assessment and treatment outcome of rosuvastatin.

## Conclusion

This study suggests that ABCG2 421 G>T polymorphism may influence the risk of development of adverse effects of rosuvastatin. Incidence of myopathy, hepatotoxicity and renal toxicity was found to be 5.3, 3.2 and 4.8 percent respectively. Due to high burden of complications of hyperlipidemia, it may be suggested that before prescribing rosuvastatin, dose may be optimized in variant T allele carriers to mitigate the risk of adverse effects which will ultimately improve patient's compliance and reduce the health burden of cardiovascular diseases.

## Supporting information

**S1. Supplemental Figures and Tables.**
(DOCX)

**S2. Gel pictures.**
(DOCX)

## Acknowledgments

We are thankful to study participants who agreed to participate and donated their blood for this research study.

## Author contributions

**Conceptualization:** Mahjabeen Sharif, Kulsoom Farhat.

**Data curation:** Mahjabeen Sharif, Kulsoom Farhat, Mudassar Noor, Ahsan Maqbool Ahmad, Dilshad Ahmed Khan.

**Formal analysis:** Mahjabeen Sharif, Kulsoom Farhat, Ahsan Maqbool Ahmad, Dilshad Ahmed Khan, Muhammad Bilal Siddique.

**Investigation:** Mahjabeen Sharif, Kulsoom Farhat, Raja Kamran Afzal.

**Methodology:** Mahjabeen Sharif, Kulsoom Farhat, Mudassar Noor, Ahsan Maqbool Ahmad, Raja Kamran Afzal, Muhammad Bilal Siddique.

**Project administration:** Mahjabeen Sharif, Kulsoom Farhat, Mudassar Noor, Muhammad Bilal Siddique.

**Software:** Dilshad Ahmed Khan, Raja Kamran Afzal.

**Supervision:** Mahjabeen Sharif, Kulsoom Farhat.

**Writing – original draft:** Mahjabeen Sharif.

**Writing – review & editing:** Mahjabeen Sharif, Mudassar Noor, Ahsan Maqbool Ahmad, Dilshad Ahmed Khan, Raja Kamran Afzal, Muhammad Bilal Siddique.

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
