## [Decision Letter · Decision Letter 0]

22 Jul 2025

PONE-D-25-29875Association of ABCG2 421G>T (rs2231142) Polymorphism with rosuvastatin induced Adverse effects in dyslipidemic patients: Implication for personalized medicine.PLOS ONE

Dear Dr. Sharif,

Thank you for submitting your manuscript to PLOS ONE. After careful consideration, we feel that it has merit but does not fully meet PLOS ONE’s publication criteria as it currently stands. Therefore, we invite you to submit a revised version of the manuscript that addresses the points raised during the review process.

Please revise and respond to each of the suggestions provided by myself the Academic Editor and Reviewer 1 below.  

We look forward to receiving your revised manuscript.

Kind regards,

James M Wright

Academic Editor

PLOS ONE

Journal Requirements: 

Additional Editor Comments:

This study provides some interesting findings regarding the adverse effects of rosuvastatin. Some revisions are required.

1. The Background should document the known magnitude of effect of rosuvastatin on LDL cholesterol the average reduction of LDL cholesterol with 10 mg of rosuvastatin is 46%. See Adams SP, Sekhon SS, Wright JM. Rosuvastatin for lowering lipids. Cochrane Database of Systematic Reviews 2014, Issue 11. Art. No.: CD010254. DOI: 10.1002/14651858.CD010254.pub2.

2. The study documents important adverse effects of rosuvastatin on muscle, liver and the kidney. These are not very low and such terms should not be used.

3. Also refrain from using well tolerated. I would judge that this study does not demonstrate that 10 mg of rosuvastatin was well tolerated.

Reviewers' comments:

Reviewer's Responses to Questions

**Comments to the Author**

1. Is the manuscript technically sound, and do the data support the conclusions?

Reviewer #1: Partly

2. Has the statistical analysis been performed appropriately and rigorously? 

Reviewer #1: Yes

3. Have the authors made all data underlying the findings in their manuscript fully available?

Reviewer #1: Yes

4. Is the manuscript presented in an intelligible fashion and written in standard English?

Reviewer #1: No

5. Review Comments to the Author

Reviewer #1: This study examines the relationship between the ABCG2 421G>T (rs2231142) polymorphism, plasma rosuvastatin levels, and the incidence of rosuvastatin-related adverse events (muscular, hepatic, and renal) in a Pakistani population. A total of 376 patients who received 10 mg of rosuvastatin daily for 12 weeks were genotyped, and their plasma drug levels, and biochemical markers were evaluated at the end of the study. The authors report a genotype-dependent increase in rosuvastatin plasma levels (TT > GT > GG) and a higher incidence of adverse events in T allele carriers. Multiple genetic models were analyzed (co-dominant, dominant, and over-dominant), all of which showed significant association, except for the over-dominant model (which was incorrectly reported as non-significant ?).

This study provides original data on an unexplored population. However, serious issues regarding methodological clarity, scientific interpretation, structure, and language quality must be addressed before the manuscript is suitable for publication.

Abstract

• The statement that “rosuvastatin 10 mg is well tolerated by most of the patients” appears contradictory, given the subsequent claim that carriers of the T allele had significantly higher plasma concentrations and increased risk of adverse effects. This undermines the conclusion and confuses the message. A more nuanced interpretation is needed.

• The recommendation that "genetic profiling should be done for dose tailoring" is too strong based on the observational design and absence of outcome validation. While the data suggest a potential association, no clinical decision-making can be inferred at this stage.

• Minor : l35 heterogenous to heterozygous

Introduction

• The introduction provides useful background on statin pharmacogenetics and the role of ABCG2 in rosuvastatin disposition. However, the section would benefit from greater conciseness and a more focused narrative.

o Several parts are redundant, particularly the repeated emphasis on genotype frequencies in different populations, which, while informative, could be condensed or moved to the discussion.

o The study objective, although implied, is not directly stated at the end of the introduction. Clearly articulating the aim e.g., "evaluate the association between ABCG2 421G>T and rosuvastatin plasma concentration and adverse events incidence in a Pakistani cohort”.

Method

• It is unclear whether a study protocol was developed a priori or if the study was prospectively registered. The absence of such information limits confidence in the prespecified nature of the endpoints and analysis plan.

• The authors should consider following a standardized reporting guideline such as STROBE for observational studies. This would help structure key methodological details more clearly.

• The manuscript does not indicate how many patients were initially screened, how many were excluded, or whether any were lost to follow-up. It is therefore impossible to assess selection bias or generalizability. The inclusion of a patient flowchart would be very helpful to clarify how the final study population of 374 patients was obtained.

• All patients received 10 mg rosuvastatin daily for 12 weeks; however, the timing of blood sampling relative to the last dose is not described. This is critical to interpreting plasma rosuvastatin concentration.

• Although logistic regression was used to assess associations between genotypes and adverse effects. The manuscript does not clarify whether covariates (e.g., age, sex, BMI, renal function, comorbidities) were included in the model. If not, how can confounding be excluded ?

Results

• Table 1: The reported BMI is relatively high, bordering on obesity. This is an important clinical parameter that should be considered in the discussion.

• Table 2: The current legend reads more like a methods detail (“Statistical analysis was performed by using an online HWE calculator”) than a description of the table contents. It would be more appropriate to move this statement to the methods section.

• Figure 1 : To improve clarity, the legend could be slightly expanded to better guide the reader’s interpretation of the figure.

• Table 3 / fig 2: Figure 2 does not add supplemental information beyond what is already presented in Table 3. It would be advisable to retain only one of the two in the main manuscript and place the other in the supplementary material.

• Table 5, the post-treatment CPK values (272.42 ± 477.8) and the reported mean change (172.22 ± 470.69) show extremely wide standard deviations, wider than the means themselves. Similar patterns are seen for other biochemical markers such as ALT and ALP. What explains this considerable variability? Could it be due to a few extreme outliers ? Additionally, this important aspect of variability is not discussed in the manuscript, despite its potential clinical and statistical implications.

• Table 7: Among the 35 TT patients, none were classified as SRM0 or SRM1 or SRM2 , only the most severe grades (SRM3 and SRM4) were observed. This unusual distribution is intriguing and raises the question of whether it is simply due to the small sample size in this group or if it reflects a distinct clinical profile associated with the TT genotype. This point warrants discussion in the manuscript.

• l338-339: There appears to be an inconsistency between the results described in the text (and in the abstract), where it is stated that "All the genetic models showed significant association with rosuvastatin adverse effects except the overdominant model", and the values presented in Table 10. This contradiction should be clarified. If the overdominant model is indeed significant, the text and abstract must be corrected accordingly. Conversely, if it was not intended to be interpreted as significant, the authors should explain why, despite the low p-value.

• The Results section, as currently presented, is overly dense. The large number of tables and figures (some of which are redundant or offer limited added value) makes the section unnecessarily dense. Only the most essential and clinically relevant results should be presented in the main manuscript, while secondary data could be moved to supplementary materials. Streamlining this section would significantly improve its readability and clarity.

Discussion

• The Discussion section requires important revision. It is too long and lacks focus, suffering from unnecessary repetition. Key findings are not sufficiently emphasized or critically examined in light of existing literature. The narrative lacks logical structure, and the various outcome domains (e.g., muscle toxicity, hepatic and renal parameters, and genetic associations) are presented in a disorganized manner. Additionally, the tone occasionally shifts toward speculation or clinical overstatement when discussing pathophysiological mechanisms or therapeutic recommendations, neither of which is fully supported by the data.

Most critically, there is no clearly defined limitations section. Major methodological weaknesses, such as lack of adjustment for confounders, small sample size in certain subgroups, and potential bias, are either overlooked or inadequately acknowledged. These issues substantially undermine the strength of the conclusions and should be transparently discussed.

Conclusion

• The conclusion overstates the implications of the findings. While the reported incidence of adverse effects is low, the recommendation to perform routine genetic testing before prescribing rosuvastatin is not sufficiently supported by the observational design of the study.

Overall

• The manuscript contains numerous formatting issues, particularly the frequent presence of double or triple spaces between words. This should be carefully corrected throughout the text.

6. PLOS authors have the option to publish the peer review history of their article (what does this mean? ). If published, this will include your full peer review and any attached files.

**Do you want your identity to be public for this peer review?** For information about this choice, including consent withdrawal, please see our Privacy Policy .

Reviewer #1: No

---

## [Author Response · Author response to Decision Letter 1]

21 Sep 2025

Respected Sir/Madam,

I am thankful to additional editors and reviewers for their input. The answers to the observations have been given against the observations. Hope the answers would suffice.

Best regards.

Point by Point Response to Additional Editors Reviewer’s Comments

Revisions/ rectifications have been made in this manuscript in the light of Additional Editor’s and Reviewer’s comments and revised text has been highlighted in red font in manuscript text.

Response to Comments of Additional Editors

Q1. The Background should document the known magnitude of effect of rosuvastatin on LDL cholesterol the average reduction of LDL cholesterol with 10 mg of rosuvastatin is 46%. See Adams SP, Sekhon SS, Wright JM. Rosuvastatin for lowering lipids. Cochrane Database of Systematic Reviews 2014, Issue 11. Art. No.: CD010254. DOI: 10.1002/14651858.CD010254.pub2.

Answer: Average reduction of LDL cholesterol with 10 mg of rosuvastatin is 46%, has been added in introduction section page 4 Line No: 73, 74. Above mentioned reference has been added in reference section (page 27, line 592, 593).

Q2. The study documents important adverse effects of rosuvastatin on muscle, liver and the kidney. These are not very low and such terms should not be used.

Answer: I have deleted the sentence “incidence of adverse effects of rosuvastatin was very low throughout the manuscript (Abstract, Discussion and conclusion section).

Q3. Also refrain from using well tolerated. I would judge that this study does not demonstrate that 10 mg of rosuvastatin was well tolerated.

Answer. Statement “Rosuvastatin is well tolerated” has been omitted in abstract, discussion, and conclusion section.

Response to Reviewer’s Comments

Abstract

Q1. The statement that “rosuvastatin 10 mg is well tolerated by most of the patients” appears contradictory, given the subsequent claim that carriers of the T allele had significantly higher plasma concentrations and increased risk of adverse effects. This undermines the conclusion and confuses the message. A more nuanced interpretation is needed.

Answer. Statement “rosuvastatin 10 mg is well tolerated by most of the patients” has been deleted in abstract portion.

Q2. The recommendation that "genetic profiling should be done for dose tailoring" is too strong based on the observational design and absence of outcome validation. While the data suggest a potential association, no clinical decision-making can be inferred at this stage.

Answer: Sentence has been rephrased as “ It is therefore suggested that genetic profiling may be done for dose tailoring to minimize the statin intolerance” ( page no.03, line no. 46, 47)

Q3.Minor: l35 heterogenous to heterozygous

Answer. Heterogenous has been replaced by heterozygous (page no.2, line No 34).

Introduction

Q1. The introduction provides useful background on statin pharmacogenetics and the role of ABCG2 in rosuvastatin disposition. However, the section would benefit from greater conciseness and a more focused narrative.

Answer. Introduction section has been revised as per directives of reviewers comments.

Q2. Several parts are redundant, particularly the repeated emphasis on genotype frequencies in different populations, which, while informative, could be condensed or moved to the discussion.

Answer. Introduction section has been rectified and genotype frequencies in different population has been moved from introduction to discussion section ( page no: 21).

Q3. The study objective, although implied, is not directly stated at the end of the introduction. Clearly articulating the aim e.g., "evaluate the association between ABCG2 421G>T and rosuvastatin plasma concentration and adverse events incidence in a Pakistani cohort”.

Answer. Study objectives and aim has been elaborated in last paragraph of introduction section ( page no. 5, 6 Line no. 103-113).

Methods

Q1. It is unclear whether a study protocol was developed a priori. The absence of such information limits confidence in the prespecified nature of the endpoints and analysis plan.

Answer. Yes. study protocol was first approved by National University of Medical Sciences ( NUMS).Then ethical approval was sought from Ethics Review Board of NUMS.

Q2. The authors should consider following a standardized reporting guideline such as STROBE for observational studies. This would help structure key methodological details more clearly.

Answer: “Study was carried out according to the STROBE guidelines” has been written in Material and Method section under the sub- heading of study protocol (page 6, 7 lines 134-140).

Q3. The manuscript does not indicate how many patients were initially screened, how many were excluded, or whether any were lost to follow-up. It is therefore impossible to assess selection bias or generalizability. The inclusion of a patient flowchart would be very helpful to clarify how the final study population of 374 patients was obtained.

Answer. Information about number of patients screened, excluded, lost to follow up has been added in material and method section under the heading study protocol (page 6, 7 line no. 134-140). Patients flow chart has been added (page no.8, figure 01).

Q4. All patients received 10 mg rosuvastatin daily for 12 weeks; however, the timing of blood sampling relative to the last dose is not described. This is critical to interpreting plasma rosuvastatin concentration.

Answer. Blood samples for Cmax at steady state concentration were taken after 12 hours of administration of last dose of rosuvastatin ( Zhang et al., 2020 ; Lee at al., 2013;) mentioned on page 7 line No.146-148.

Reference of blood timings after last dosing of rosuvastatin

1. Zhang, D., Ding, Y., Wang, X. et al. Effects of ABCG2 and SLCO1B1 gene variants on inflammation markers in patients with hypercholesterolemia and diabetes mellitus treated with rosuvastatin. Eur J Clin Pharmacol 76, 939–946 (2020).

2. Lee HK, Hu M, Lui SSh, Ho CS, Wong CK, Tomlinson B. Effects of polymorphisms in ABCG2, SLCO1B1, SLC10A1 and CYP2C9/19 on plasma concentrations of rosuvastatin and lipid response in Chinese patients. Pharmacogenomics. 2013 Aug;14(11):1283-94.

Q5. Although logistic regression was used to assess associations between genotypes and adverse effects. The manuscript does not clarify whether covariates (e.g., age, sex, BMI, renal function, comorbidities) were included in the model. If not, how can confounding be excluded ?

Answer. Binary logistic regression analysis was applied and potential confounders like age, gender, BMI, WC, ethnicities, CPK, LFTs, RFTs were adjusted (page no. 20, line no.441, 443).

Results

Q1. Table 1: The reported BMI is relatively high, bordering on obesity. This is an important clinical parameter that should be considered in the discussion.

Answer. Description of high BMI amongst enrolled hyperlipidemic patients has been added in discussion section ( Page no: 21).

Q2. Table 2: The current legend reads more like a methods detail (“Statistical analysis was performed by using an online HWE calculator”) than a description of the table contents. It would be more appropriate to move this statement to the methods section.

Answer. Statement “Statistical analysis was performed by using an online HWE calculator” has been moved from description of table content to method section under the sub-heading of Statistical Analysis (page 12, line No. 268-270).

Q3. Figure 1 : To improve clarity, the legend could be slightly expanded to better guide the reader’s interpretation of the figure.

Answer: Legend of figure 1 (now supplementary figure S1) has been elaborated.

Q4. Table 3 / fig 2: Figure 2 does not add supplemental information beyond what is already presented in Table 3. It would be advisable to retain only one of the two in the main manuscript and place the other in the supplementary material.

Answer. Table 3 has been retained in the manuscript and Figure 2 has been placed in supplementary figure S2.

Q5. Table 5, the post-treatment CPK values (272.42 ± 477.8) and the reported mean change (172.22 ± 470.69) show extremely wide standard deviations, wider than the means themselves. Similar patterns are seen for other biochemical markers such as ALT and ALP. What explains this considerable variability? Could it be due to a few extreme outliers ? Additionally, this important aspect of variability is not discussed in the manuscript, despite its potential clinical and statistical implications.

Justification

Wide standard deviations in post treatment CPK levels has been observed as some patients with TT and GT genotypes, have ten times higher levels of post treatment CPK levels which falls in SRM 4 group. These higher levels of CPK >10xUNL--<50XULN produced severe myopathy and is the most important finding of this study which might produce wide standard deviation. This inter individual variability in three genotype groups of ABCG2 421G>T has been mentioned in discussion section (page 22, line no. 478-485 page 23 line no. 489-494).

Wide standard deviations in post treatment values of serum ALT and AST were observed as some patients carrying T mutant allele (TT and GT genotypes) have raised serum ALT and ALP levels >2.5-5.0XULN, which is important clinical finding of this study (grade 2 liver injury). These extreme values may produce wide SD values. This considerable variability in post treatment values of ALT and AST was observed in patients having TT and GT genotypes has been mentioned in discussion section (page no. 23, 24& line no. 511-516).

Q 6. Table 7: Among the 35 TT patients, none were classified as SRM0 or SRM1 or SRM2 , only the most severe grades (SRM3 and SRM4) were observed. This unusual distribution is intriguing and raises the question of whether it is simply due to the small sample size in this group or if it reflects a distinct clinical profile associated with the TT genotype. This point warrants discussion in the manuscript.

Answer:

It might be due to the fact that patients carrying TT genotypes have raised plasma rosuvastatin levels due to decreased function of efflux BCRP transporter which ultimately raises the levels of rosuvastatin in muscles, predisposing to increased risk of muscular adverse effects. So most of the patients with TT genotype have SRM 3 and SRM4 grade myopathy

15 out of 35 patients did not experience muscular symptoms but none of the patients with TT genotypes were classified as SRM0 , SRM1 or SRM2 grade muscular symptoms. It might be due to small sample size of patients carrying TT genotypes, which is also a novel finding of this study on Pakistani population and has been explained in discussion section (page no. 22, line no. 478-485 & Page no. 22-23, line no. 489-494).

Moreover our observed genotypes count was consistent with expected genotype count, showing that our results are in agreement with Hardy Weinberg Equilibrium.

Q7. l338-339: There appears to be an inconsistency between the results described in the text (and in the abstract), where it is stated that "All the genetic models showed significant association with rosuvastatin adverse effects except the overdominant model", and the values presented in Table 10. This contradiction should be clarified. If the overdominant model is indeed significant, the text and abstract must be corrected accordingly. Conversely, if it was not intended to be interpreted as significant, the authors should explain why, despite the low p-value.

Answer: There is mistake at my end as binary logistic regression analysis has shown that “All the genetic models showed significant association with rosuvastatin adverse effects “. I have rectified it in abstract section (page no. 2-3 line no. 40-44) and result section under the sub-heading of Genetic Model Association of rs2231142 with rosuvastatin adverse effects

(page no. 20, line no.442-443).

Q 8. The Results section, as currently presented, is overly dense. The large number of tables and figures (some of which are redundant or offer limited added value) makes the section unnecessarily dense. Only the most essential and clinically relevant results should be presented in the main manuscript, while secondary data could be moved to supplementary materials. Streamlining this section would significantly improve its readability and clarity.

Answer: Whole Result Section has been revised and stream lined. Only most important and clinically relevant data has been presented in manuscript and the secondary data has been moved to supplementary files.

Discussion

Q1.The Discussion section requires important revision. It is too long and lacks focus, suffering from unnecessary repetition. Key findings are not sufficiently emphasized or critically examined in light of existing literature. The narrative lacks logical structure, and the various outcome domains (e.g., muscle toxicity, hepatic and renal parameters, and genetic associations) are presented in a disorganized manner. Additionally, the tone occasionally shifts toward speculation or clinical overstatement when discussing pathophysiological mechanisms or therapeutic recommendations, neither of which is fully supported by the data.

Answer: Discussion section has been revised and most of the unnecessary repetition has been deleted. Clinical overstatement about pathophysiological mechanisms and therapeutic recommendations has been revised and rephrased (page no. 25, lines no. 543-551).

Q. Critically, there is no clearly defined limitations section. Major methodological weaknesses, such as lack of adjustment for confounders, small sample size in certain subgroups, and potential bias, are either overlooked or inadequately acknowledged. These issues substantially undermine the strength of the conclusions and should be transparently discussed.

Answer. Study limitations have been added in last paragraph of discussion section (page 25 line no. 552-557).

Lack of adjustment for confounders

Binary logistic regression analysis was applied and potential confounders like age, gender, BMI, waist circumference, ethnicities, CPK, LFTs, RFTs were adjusted to assess the significant association of different genotypes with rosuvastatin adverse effects (page 20, line no. 441-443).

Justification of small sample size in certain groups:

Genotyping of 400 study patients was carried out and out of 400 patients 35 patients have TT genotypes and our observed genotypes were consistent with expected genotypes showing that our results are in agreement with Hardy Weinberg Equilibrium. Small sample size of TT genotype has shown that frequency of TT (mutant homozygous) genotype in Pakistani population is low compared with GT and GG genotypes. This is a novel finding in Pakistani population.

Conclusion

The conclusion overstates the implications of the findings. While the reported incidence of adverse effects is low, the recommendation to perform routine genetic testing before prescribing rosuvastatin is not sufficiently supported by the observational design of the study.

Answer: Conclusion section has been revised as per directives of reviewer (page 26, 560-565).

Overall

• The manuscript contains numerous formatting issues, particularly the frequent presence of double or triple spaces between words. This should be carefully corrected throughout the text.

Answer: Formatting issues, double and triple spaces between words and other grammatical mistakes have been rectified throughout the text of manuscript.

---

## [Editor Report · Decision Letter 1]

1 Oct 2025

Association of ABCG2 421G>T (rs2231142) Polymorphism with rosuvastatin induced Adverse effects in dyslipidemic patients: Implication for personalized medicine.

PONE-D-25-29875R1

Dear Dr. Sharif,

We’re pleased to inform you that your manuscript has been judged scientifically suitable for publication and will be formally accepted for publication once it meets all outstanding technical requirements.

Kind regards,

James M Wright

Academic Editor

PLOS ONE
---

## [Editor Report · Acceptance letter]

PONE-D-25-29875R1

PLOS ONE

Dear Dr. Sharif,

I'm pleased to inform you that your manuscript has been deemed suitable for publication in PLOS ONE. Congratulations! Your manuscript is now being handed over to our production team.

Kind regards,

on behalf of

Professor James M Wright

Academic Editor

PLOS ONE